# An examination of autonomic and facial responses to prototypical facial emotion expressions in psychopathy

**Philip Deming**[1,2]*, **Hedwig Eisenbarth**[3], **Odile Rodrik**[1,2], **Shelby S. Weaver**[1,2], **Kent A. Kiehl**[4,5], **Michael Koenigs**[1,2]

**1** Department of Psychology, University of Wisconsin-Madison, Madison, Wisconsin, United States of America, **2** Department of Psychiatry, University of Wisconsin-Madison, Madison, Wisconsin, United States of America, **3** School of Psychology, Victoria University of Wellington, Wellington, New Zealand, **4** The Mind Research Network and Lovelace Biomedical, Albuquerque, New Mexico, United States of America, **5** Department of Psychology, University of New Mexico, Albuquerque, New Mexico, United States of America

* pdeming@wisc.edu

**Data Availability Statement:** The data are available through the figshare repository at the following DOI: 10.6084/m9.figshare.20089352.

## Abstract

Meta-analyses have found that people high in psychopathy categorize (or "recognize") others' prototypical facial emotion expressions with reduced accuracy. However, these have been contested with remaining questions regarding the strength, specificity, and mechanisms of this ability in psychopathy. In addition, few studies have tested holistically whether psychopathy is related to reduced facial mimicry or autonomic arousal in response to others' dynamic facial expressions. Therefore, the current study presented 6 s videos of a target person making prototypical emotion expressions (anger, fear, disgust, sadness, joy, and neutral) to $N = 88$ incarcerated adult males while recording facial electromyography, skin conductance response (SCR), and heart rate. Participants identified the emotion category and rated the valence and intensity of the target person's emotion. Psychopathy was assessed via the Psychopathy Checklist-Revised (PCL-R). We predicted that overall PCL-R scores and scores for the interpersonal/affective traits, in particular, would be related to reduced emotion categorization accuracy, valence ratings, intensity ratings, facial mimicry, SCR amplitude, and cardiac deceleration in response to the prototypical facial emotion expressions. In contrast to our hypotheses, PCL-R scores were unrelated to emotion categorization accuracy, valence ratings, and intensity ratings. Stimuli failed to elicit facial mimicry from the full sample, which does not allow drawing conclusions about the relationship between psychopathy and facial mimicry. However, participants displayed general autonomic arousal responses, but not to prototypical emotion expressions per se. PCL-R scores were also unrelated to SCR and cardiac deceleration. These findings failed to identify aberrant behavioral and physiological responses to prototypical facial emotion expressions in relation to psychopathy.

**Funding:** The author(s) received no specific funding for this work.

**Competing interests:** The authors have declared that no competing interests exist.

## Introduction

Psychopathy is a personality disorder characterized by a deceitful interpersonal style, callousness, and impulsivity [1]. In particular, a diminished capacity to categorize (or "recognize") prototypical facial emotion expressions is thought to be associated with the deceitful interpersonal and callous affective features of psychopathy [2]. Indeed, when confronted with a variety of prototypical facial emotion expressions, including the emotion categories of joy, fear, sadness, surprise, and anger, people high in psychopathy display reduced categorization accuracy [2–4]. However, questions remain regarding the strength, specificity, and mechanisms of facial expression processing in psychopathy. Clarifying the physiological mechanisms that are disrupted during prototypical facial emotion expression processing could have implications for identification and treatment of the disorder.

Facial mimicry has been proposed as one physiological mechanism supporting the process of categorizing or recognizing others' emotions from facial expressions. Embodied simulation accounts posit that spontaneous mimicry of another's facial expression evokes neural representations of the correspondent emotion category, providing the perceiver access to approximations of the other's emotional state [5,6]. Indeed, healthy individuals spontaneously mimic others' facial expressions [7,8] and categorize others' prototypical facial emotion expressions less accurately [9] and more slowly [10] when facial mimicry is blocked. Moreover, reduced facial mimicry is associated with low capacity for empathy, a key characteristic of psychopathy [11]. While these findings collectively suggest the possibility of a facial mimicry deficit in psychopathy, such data are lacking. Support for this hypothesis can be found in one study that found reduced mimicry of the *zygomaticus major* and *corrugator supercilii* muscles, measured via electromyography (EMG), among juveniles with callous-unemotional traits, thought to be a precursor to psychopathy in adulthood [12]. On the contrary, a study of *corrugator supercilii* muscle activity among adults high in psychopathy revealed no facial mimicry deficit [13]. Thus, the hypothesis that psychopathy is related to reduced spontaneous facial mimicry has been largely unexplored and merits further investigation of additional facial muscles. In the current study, we measure the activity of four facial muscles involved in prototypical facial emotion expressions to test this hypothesis.

Additionally, autonomic arousal may be diminished when individuals high in psychopathy perceive another's facial expression. Healthy individuals display autonomic arousal as indexed by skin conductance response (SCR) [14], cardiac deceleration [15], and pupil dilation [16,17] when perceiving another's prototypical facial emotion expression. Studies investigating the relationship between psychopathy and autonomic arousal in response to prototypical facial emotion expressions have focused on pupil dilation and yielded mixed findings. Two studies found a negative relationship between interpersonal/affective psychopathic traits and pupil response [18,19], and two studies found no relationship between psychopathic traits and pupil response [20,21]. Youth with psychopathic traits have shown reduced SCR to prototypical fear faces but not prototypical anger faces [22]. Moreover, a broader literature has associated psychopathy, particularly the interpersonal/affective psychopathic traits, with reduced autonomic arousal to a variety of emotional stimuli, including witnessing another person receiving electrical shocks [23,24], imagining emotional scenarios [25,26], and perceiving emotional sounds [27] and pictures [28]. However, these findings are also mixed [4,29]. Thus, further research is needed to characterize the relationship between psychopathy and autonomic arousal (e.g., SCR and cardiac deceleration) in response to prototypical facial emotion expressions.

Importantly, there is reason to scrutinize the assumption that any person, irrespective of psychopathy, can accurately "read" emotions from faces. This assumption stemmed from cross-cultural research that claimed to identify universal "basic emotions," which we call

"emotion categories," that correspond to unique facial configurations, which we refer to as "prototypical emotion expressions" [30]. Accumulating evidence suggests that people do not reliably smile when happy, scowl when angry, or frown when sad [31,32]. There is not a one-to-one mapping between a person's emotional state and facial configuration. Moreover, facial expressions also appear to convey social information [33] and information about two dimensions of a person's affect, namely valence and arousal [31,34]. To characterize the callousness of people with psychopathy, a large body of research has examined how they perceive emotion categories from faces [2–4,35,36], but only one study has examined how they perceive affect from faces [37]. Women high in psychopathy perceived another person's prototypical joy expressions as less positive (i.e., more neutral in valence), compared to women low in psychopathy. In the current study, we attempted to replicate and extend this finding. The shifting science of how healthy people use facial configurations to express and perceive affect and emotion could lead to new insights into the callous lack of empathy in psychopathy. Identifying aberrant mechanisms in psychopathy could also advance our understanding of how these mechanisms support healthy individuals' perceptions of affect and emotion in facial expressions.

The current study tested whether psychopathy is associated with alterations in behavioral and physiological responses to prototypical facial emotion expressions. We hypothesized behavioral and physiological alterations in relation to the total construct of psychopathy and to the interpersonal/affective features of psychopathy (e.g., lack of empathy, shallow affect), in particular, but not to the lifestyle/antisocial features of psychopathy (e.g., impulsivity, irresponsibility). Specifically, we predicted that psychopathy would be related to diminished categorization accuracy for all presented prototypical emotion expressions, given prior literature [2,4]. We predicted that psychopathy would be related to more neutral valence ratings of joy (but not negative emotions) and lower intensity ratings of all emotions [37]. We also predicted that psychopathy would be related to reduced mimicry (i.e., EMG activity of four facial muscles), based on previous findings that reduced facial mimicry is associated with low capacity for empathy [11]. Finally, we predicted psychopathy would be related to reduced autonomic arousal (i.e., SCR and cardiac deceleration) in response to others' prototypical facial emotion expressions.

## Materials and methods

### Participants

Male incarcerated individuals between the ages of 18 and 55 were recruited from a medium-security correctional facility in Wisconsin. Included participants had no history of psychosis, bipolar disorder, epilepsy or stroke, were not currently using antipsychotic, antianxiety, tricyclic antidepressant, or mood stabilizer medications, had no history of head injury with loss of consciousness >30 minutes, attained >4th grade English reading level and >70 IQ, and had intact audition and vision. Eighty-eight participants met inclusion criteria and completed the current study. All participants provided written informed consent. Participants were informed that their participation was voluntary and would not affect their institutional status. The study was approved by the Health Sciences Institutional Review Board at the University of Wisconsin-Madison (ID 2016–1073).

### Assessments

Psychopathy was assessed with the Psychopathy Checklist-Revised (PCL-R) [1]. The twenty items were rated on a 0–2 scale, based on information obtained during a 60–90 minute interview and institutional file review. Scores for PCL-R Factor 1 (interpersonal/affective traits) and

Factor 2 (lifestyle/antisocial traits) were derived according to published guidelines [1]. Anxiety was assessed via the Welsh Anxiety Inventory (WAI), a self-report measure with 39 true-false items [38]. IQ was estimated from the Wechsler Adult Intelligence Scale 3$^{rd}$ Ed. [39] vocabulary and matrix reasoning subscales. Lifetime substance use disorder diagnoses (for any substance) were determined using the Structured Clinical Interview for the DSM-IV [40]. See participant characteristics in Table 1.

## Prototypical facial expression task

Participants viewed videos from the Amsterdam Dynamic Facial Expression Set, a validated stimulus set featuring actors making prototypical facial displays of emotion [41]. See Fig 1 for examples of stimuli and the trial time course. Each 6 s video consisted of a forward-facing white actor (whom we call the "target person") displaying a prototypical neutral facial expression at stimulus onset, beginning to form a prototypical emotion expression 1–2 s after stimulus onset, reaching the peak of the expression 3–4 s after stimulus onset, and maintaining the expression until stimulus offset. Videos from 10 target persons (five female) were presented, with each target person posing five prototypical emotion expressions (anger, disgust, fear, sadness, joy) and one neutral expression, resulting in 60 total trials.

Following each video, participants responded to three questions. First, participants identified which emotion the target person felt from one of six options ("Anger", "Disgust", "Fear", "Joy", "Sadness", or "No Emotion"). Next, participants rated the valence of the target person's emotion on a seven-point Likert scale (-3 to 3, with anchors at -3 for "very bad", 0 for "neutral", and 3 for "very good") and the intensity of the target person's emotion on a seven-point Likert scale (0 to 6, with anchors at 0 for "not at all intense" and 6 for "very intense"). Rating screens were self-timed ($M$ = 3.8 s per rating screen, $SD$ = 0.9). Participants selected responses using the computer keyboard with their right hand. A 1 s fixation cross was displayed before each stimulus onset. Participants were instructed to watch the videos and respond to the questions after each video. No instructions regarding mimicry were given.

## Physiological data acquisition

The BIOPAC MP160 (BIOPAC Systems Inc., Goleta, CA, USA) physiological monitoring system was used to acquire EMG, skin conductance, and heart rate at a sampling rate of 2,000 Hz. Four pairs of Ag/AgCl EMG electrodes (4 mm recording diameter, filled with BIOPAC electrode gel, GEL100) were attached over four facial muscles on the right side of the face according to published guidelines [42,43]: *corrugator supercilii* (brow lowerer), *levator labii superioris*

Table 1. Participant characteristics ($N$ = 88).

| Measure | $M$ *(SD)* | Range |
|---|---|---|
| PCL-R Total | 23.8 (7.7) | 6.7–34.7 |
| PCL-R Factor 1 | 9.1 (3.0) | 0.0–15.0 |
| PCL-R Factor 2 (*n* = 81) | 12.6 (4.6) | 1.3–20.0 |
| Age (Years) | 38.4 (7.6) | 20.0–55.0 |
| Welsh Anxiety (*n* = 81) | 12.0 (9.6) | 0.0–39.0 |
| IQ | 98.6 (11.8) | 74.0–124.0 |
| Measure | % | |
| Race/Ethnicity (White) | 44.3 | |
| Substance Use Disorder | 80.7 | |

*Note. PCL-R = Psychopathy Checklist-Revised* [1].

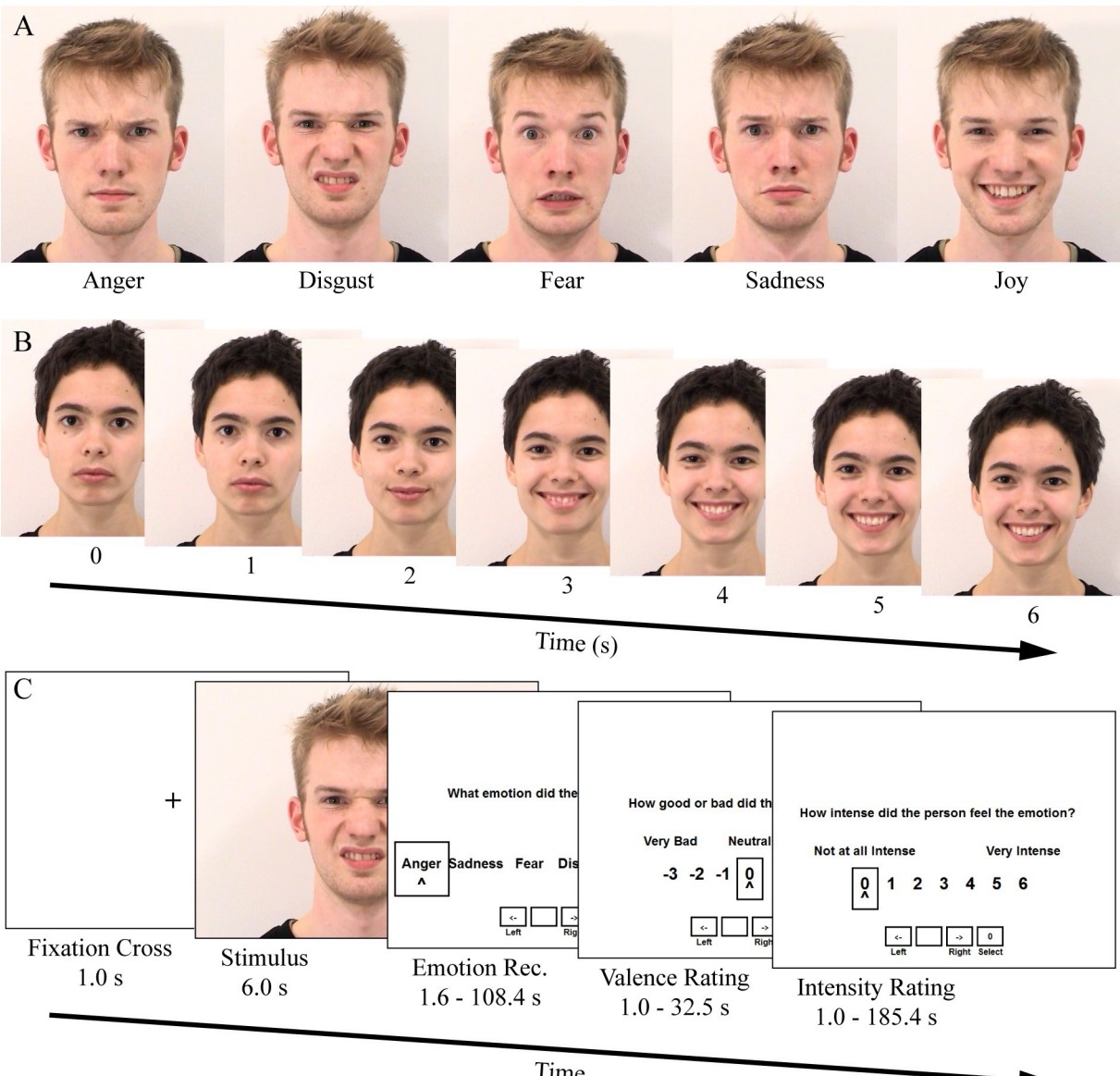

**Fig 1. Prototypical facial expression task.** A) Examples of each prototypical facial expression of emotion from the Amsterdam Dynamic Facial Expression Set, captured at the peak of each expression (about 3–4 s after stimulus onset). B) Time course of an example video. The target person transitioned from a neutral facial display to an emotional expression around 1–2 s after stimulus onset. C) Time course of a single trial. A fixation cross was followed by the stimulus, then by response screens for emotion categorization, valence rating, and intensity rating of the target person's facial emotion expression.

(nose wrinkler), *zygomaticus major* (lip corner puller), and *depressor anguli oris* (lip corner depressor). Each muscle site was first cleaned with rubbing alcohol and BIOPAC abrasive gel to ensure electrode impedance was <25 kΩ. Skin conductance was acquired via two Ag/AgCl electrodes (11 mm recording diameter, filled with BIOPAC isotonic electrode paste, GEL101A) on the thenar/hypothenar surface of the left hand. Heart rate was monitored via pulse plethysmography attached to the tip of the left index finger.

Participants' eye gaze to different regions of the target person's face were measured with an eye-tracking device. However, calibration issues due to eyeglasses, participant movement, and

limitations in controlling the lighting in the prison setting led to poor data quality for many participants. Details about eye-tracking are presented in the supporting information (S1 File).

### Physiological data processing

EMG data from each muscle were processed and analyzed independently using the 'biosignalEMG' package in R [44]. Filters (60 Hz notch, 10 Hz high pass, and 500 Hz low pass) were applied to EMG time series according to published guidelines [45]. A 60 Hz notch filter was applied because a fast Fourier transform showed a peak at 60 Hz due to electrical noise. EMG signals were then integrated with 200 ms time constant and z-scored within subjects. Facial mimicry was operationalized as the average z-score 2–5 s after stimulus onset (the time window recommended by reviewers), given that the target person began making a prototypical facial expression 1–2 s after stimulus onset and reached the peak 3–4 s after stimulus onset. According to prior work, the EMG response to a target person's facial expression can be detected within 1 s [8]. To control for baseline EMG activity, average z-scores 500 ms before stimulus onset were subtracted from average z-scores 2–5 s after stimulus onset. Participants were instructed after the task to deliberately move each muscle to ensure that each muscle's activity was measured. Data from deliberate muscle movements are in the supporting information (S1 File).

Skin conductance data was processed in MATLAB [46] using the PsPM toolbox (github.com/bachlab/pspm). Stimulus-evoked SCR amplitude was estimated via dynamic causal modeling, which allows SCR onset to vary within a specified time window (i.e., stimulus presentation), with a canonical SCR function two trials at a time. For each trial, SCRs were modeled separately during the 6 s stimulus presentation and the self-timed rating screen presentation, to account for SCRs possibly resulting from physical motion during rating screen presentation (i.e., button presses). Prior to dynamic causal modeling, artifacts were identified visually and replaced with interpolated values for 25 participants, and a standard high pass filter (cut-off frequency = 0.0159 Hz) was applied. SCR amplitudes were square root-transformed.

Cardiac deceleration was quantified in R [47] as the maximum heart rate reduction 2–5 s after stimulus onset, the window capturing the beginning and peak of the target person's prototypical facial expression. This method for quantifying cardiac deceleration has been described previously [48,49]. Heart beats were identified and inter-beat intervals (IBIs) computed using an in-house heart beat detection package. IBIs were then converted to heart rate in beats per minute and averaged into 500 ms bins. Baseline heart rate, the average heart rate 500 ms before stimulus onset, was subtracted from each time series. Finally, cardiac deceleration was calculated as the maximum heart rate reduction 2–5 s after onset of each trial.

### Multiple imputations

In total, 33 out of 88 participants (37.5%) had incomplete data. We therefore imputed missing data to avoid potential bias in the analyses. Ten imputed data sets were derived from a bootstrapped expectation maximization algorithm [the 'amelia' package in R; 50]. All independent and dependent variables were included in the multiple imputation algorithm. Statistical inference was made based on analyses pooled across the ten imputed data sets (using the 'mitml' package in R). The imputed data allowed for the inclusion of participants who were missing data for the following reasons: participant failed to complete the WAI ($n = 7$), omitted items prevented calculation of PCL-R Factor 2 score ($n = 7$), facial hair prevented EMG measurement of *depressor* activity ($n = 22$), artifactual noise prevented observation of SCRs ($n = 1$) and

heartbeats ($n$ = 1), and irregular heartbeat prevented observation of reliable stimulus-evoked cardiac deceleration ($n$ = 1).

### Data analysis

We first tested responses to the task across participants. All models were first collapsed across emotion categories (anger, disgust, fear, sadness, and joy), then separate models were run for each emotion category. To determine whether participants performed the emotion categorization task more accurately than chance, one-sample t-tests were computed with μ = .167. For the other dependent variables, we contrasted responses to prototypical facial emotion expressions with responses to neutral trials (as recommended by reviewers), in which the target person's facial muscles did not move, to control for general responses to faces. Linear mixed effects models with expression (i.e., prototypical expression vs. neutral) as a within-subjects factor examined participants' valence ratings, intensity ratings, facial mimicry, SCR, and cardiac deceleration. Unexpectedly, these models did not reveal greater facial mimicry or autonomic arousal to prototypical emotion expressions relative to neutral trials. We therefore ran additional general linear models without contrasting against neutral trials as manipulation checks.

Next, we tested the relationship between psychopathy and responses to the task. General linear models tested the relationship between PCL-R Total scores and emotion categorization accuracy. For the other dependent variables, PCL-R Total scores were added to the linear mixed effects models as a between-subjects factor. The interaction between PCL-R Total scores and expression (i.e., prototypical vs. neutral) was the effect of interest. We repeated this process to test relationships with PCL-R Factor 1 and Factor 2, with each model controlling for the other factor. For each dependent variable, Bonferroni-Holm correction was applied to the emotion category-specific tests to ensure $p_{FWE} < .050$. The number of emotion category-specific tests for facial mimicry was minimized by testing only the muscle(s) critical to each prototypical emotion expression [51]: *corrugator* for anger, *levator* for disgust, *corrugator* and *zygomaticus* for fear, *depressor* for sadness, and *zygomaticus* for joy. See supporting information (S1 File) for details.

Covariates of age, race (a dichotomous variable coded white or non-white), and WAI anxiety scores were mean-centered and included in all models for the following reasons. Aging is known to affect SCR [52], as well as heart rate response to affective stimuli [53]. People appear to mimic facial expressions of emotion to a greater extent when the other person is of the same race or in-group [54,55]. Lastly, anxiety has been found to affect SCR [56] and facial EMG response [57,58] to affective stimuli.

According to power analyses, the above tests had 80% power to detect a small effect size [$0.02 < f^2 < 0.15$; 59].

## Results

### Emotion categorization

Participants categorized the prototypical emotion expressions (collapsed across categories) with accuracy better than chance (Table 2), $t(87) = 97.90$, $p < .001$. Categorization accuracy was better than chance for all emotion categories: anger $t(87) = 40.94$, $p_{FWE} < .001$; disgust, $t(87) = 29.47$, $p_{FWE} < .001$; fear, $t(87) = 96.26$, $p_{FWE} < .001$; sadness, $t(87) = 61.36$, $p_{FWE} < .001$; joy, $t(87) = 156.00$, $p_{FWE} < .001$; neutral, $t(87) = 43.90$, $p_{FWE} < .001$.

Contrary to predictions, PCL-R Total scores were unrelated to emotion categorization accuracy across emotions, $b$ = -0.08, 95% CI [-0.18, 0.02], $F(1, 82.0) = 2.43$, $p = .123$. Nor were PCL-R Total scores related to categorization accuracy for specific emotion categories: anger, $b$

**Table 2. Task performance across all participants ($N$ = 88).**

| Performance Measure | | All Emotion Categories† | Emotion Categories | | | | | Neutral |
|---|---|---|---|---|---|---|---|---|
| | | | Anger | Disgust | Fear | Sadness | Joy | |
| **Emotion Categorization** | % | 90.4 | 84.4 | 80.1 | 95.7 | 93.4 | 98.5 | 89.9 |
| **Valence Rating** | *M* | -1.0 | -1.6 | -1.6 | -1.9 | -1.8 | 2.0 | 0.0 |
| | *SD* | 0.4 | 0.6 | 0.7 | 0.7 | 0.6 | 0.6 | 0.2 |
| **Intensity Rating** | *M* | 3.5 | 3.3 | 3.4 | 3.9 | 3.5 | 3.6 | 0.4 |
| | *SD* | 0.9 | 1.1 | 1.0 | 1.1 | 1.0 | 1.3 | 0.6 |

Note. Valence ratings were made on a seven-point scale from -3 (very bad) to 3 (very good). Intensity ratings were made on a seven point scale from 0 (not at all intense) to 6 (very intense).

† Responses were collapsed across emotion categories except neutral.

= -0.05, 95% CI [-0.09, 0.00], $F(1, 81.2)$ = 3.96; disgust, $b$ = 0.00, 95% CI [-0.06, 0.06], $F(1, 81.8)$ = 0.00; fear, $b$ = -0.01, 95% CI [-0.03, 0.01], $F(1, 81.9)$ = 0.92; sadness, $b$ = -0.02, 95% CI [-0.05, 0.02], $F(1, 81.9)$ = 0.90; joy, $b$ = -0.01, 95% CI [-0.02, 0.01], $F(1, 82.0)$ = 1.37; and neutral, $b$ = 0.02, 95% CI [-0.03, 0.07], $F(1, 81.4)$ = 0.62; all $p_{FWE}$ > .299.

PCL-R Factor 1 scores were also unrelated to emotion categorization accuracy across emotions, $b$ = -0.20, 95% CI [-0.55, 0.15], $F(1, 80.6)$ = 1.28, $p$ = .261, and for specific emotion categories: anger, $b$ = -0.02, 95% CI [-0.17, 0.13], $F(1, 80.5)$ = 0.07; disgust, $b$ = -0.03, 95% CI [-0.23, 0.18], $F(1, 80.9)$ = 0.07; fear, $b$ = -0.03, 95% CI [-0.11, 0.04], $F(1, 81.0)$ = 0.81; sadness, $b$ = -0.12, 95% CI [-0.24, -0.01], $F(1, 80.8)$ = 4.46; joy, $b$ = 0.00, 95% CI [-0.05, 0.05], $F(1, 80.8)$ = 0.00; and neutral, $b$ = -0.10, 95% CI [-0.25, 0.06], $F(1, 80.1)$ = 1.44; all $p_{FWE}$ > .227. PCL-R Factor 2 scores were similarly unrelated to emotion categorization accuracy across emotions, $b$ = 0.02, 95% CI [-0.22, 0.25], $F(1, 79.7)$ = 0.02, $p$ = .889, and for specific emotion categories: anger, $b$ = -0.05, 95% CI [-0.15, 0.06], $F(1, 79.4)$ = 0.82; disgust, $b$ = 0.02, 95% CI [-0.12, 0.15], $F(1, 80.5)$ = 0.04; fear, $b$ = 0.01, 95% CI [-0.04, 0.06], $F(1, 80.8)$ = 0.16; sadness, $b$ = 0.06, 95% CI [-0.02, 0.14], $F(1, 80.4)$ = 2.25; joy, $b$ = -0.02, 95% CI [-0.05, 0.02], $F(1, 80.4)$ = 0.78; and neutral, $b$ = 0.09, 95% CI [-0.02, 0.20], $F(1, 78.1)$ = 2.26; all $p_{FWE}$ > .653.

## Valence ratings

Compared to neutral trials, participants rated the prototypical emotion expressions (collapsed across categories) as significantly more negative (Table 2), $b$ = -0.94, 95% CI [-1.03, -0.85], $F(1, 82.1)$ = 427.12, $p$ < .001. This result was likely driven by the two thirds of trials portraying negative emotion categories (i.e., anger, disgust, fear, and sadness). Participants rated joy trials as more positive than they rated neutral trials, $b$ = 2.05, 95% CI [1.92, 2.18], $F(1, 82.0)$ = 1021.38, $p_{FWE}$ < .001. Participants rated anger, $b$ = -1.61, 95% CI [-1.73, -1.48], $F(1, 82.1)$ = 645.57, $p_{FWE}$ < .001, disgust, $b$ = -1.53, 95% CI [-1.67, -1.38], $F(1, 82.1)$ = 422.55, $p_{FWE}$ < .001, fear, $b$ = -1.84, 95% CI [-1.97, -1.70], $F(1, 82.1)$ = 722.53, $p_{FWE}$ < .001, and sadness trials, $b$ = -1.78, 95% CI [-1.90, -1.66], $F(1, 82.1)$ = 917.24, $p_{FWE}$ < .001, as more negative than they rated neutral trials.

PCL-R Total scores were unrelated to valence ratings of prototypical emotion expressions (collapsed across categories) relative to neutral trials, $b$ = 0.00, 95% CI [-0.01, 0.01], $F(1, 80.0)$ = 0.01, $p$ = .937. Contrary to predictions, PCL-R Total scores were unrelated to valence ratings for joy relative to neutral trials, $b$ = -0.01, 95% CI [-0.03, 0.01], $F(1, 82.1)$ = 1.85, $p_{FWE}$ = .885. PCL-R Total was also unrelated to valence ratings for any other specific emotion relative to neutral trials: anger, $b$ = 0.01, 95% CI [-0.01, 0.02], $F(1, 82.1)$ = 0.30; disgust, $b$ = 0.01, 95% CI

[-0.01, 0.03], $F(1, 82.1) = 0.65$; fear, $b = 0.00$, 95% CI [-0.02, 0.02], $F(1, 82.1) = 0.04$; sadness, $b = 0.00$, 95% CI [-0.01, 0.02], $F(1, 82.1) = 0.16$; all $p_{FWE} > .834$.

PCL-R Factor 1 scores were also unrelated to valence ratings of prototypical emotion expressions (collapsed across categories) relative to neutral trials, $b = 0.01$, 95% CI [-0.03, 0.05], $F(1, 77.9) = 0.08$, $p = .775$, and of specific emotion categories relative to neutral trials: anger, $b = 0.01$, 95% CI [-0.05, 0.07], $F(1, 82.0) = 0.08$; disgust, $b = 0.00$, 95% CI [-0.07, 0.07], $F(1, 78.0) = 0.00$; fear, $b = -0.01$, 95% CI [-0.07, 0.05], $F(1, 78.0) = 0.11$; sadness, $b = 0.02$, 95% CI [-0.03, 0.08], $F(1, 77.7) = 0.71$; and joy, $b = 0.01$, 95% CI [-0.05, 0.07], $F(1, 77.7) = 0.13$; all $p_{FWE} > .983$. PCL-R Factor 2 scores were similarly unrelated to valence ratings of prototypical emotion expressions (collapsed across categories) relative to neutral trials, $b = 0.00$, 95% CI [-0.03, 0.02], $F(1, 77.6) = 0.10$, $p = .756$, and of specific emotion categories relative to neutral trials: anger, $b = 0.00$, 95% CI [-0.03, 0.04], $F(1, 81.9) = 0.01$; disgust, $b = 0.01$, 95% CI [-0.03, 0.06], $F(1, 77.9) = 0.32$; fear, $b = 0.00$, 95% CI [-0.04, 0.04], $F(1, 78.0) = 0.00$; sadness, $b = -0.01$, 95% CI [-0.05, 0.02], $F(1, 77.2) = 0.42$; and joy, $b = -0.03$, 95% CI [-0.06, 0.01], $F(1, 77.3) = 1.95$; all $p_{FWE} > .834$.

## Intensity ratings

Participants rated the prototypical emotion expressions (collapsed across categories) as more intense than neutral trials (Table 2), $b = 3.11$, 95% CI [2.90, 3.33], $F(1, 82.0) = 825.30$, $p < .001$. This pattern was consistent for each specific emotion relative to neutral trials: anger, $b = 2.88$, 95% CI [2.63, 3.12], $F(1, 82.0) = 542.38$; disgust, $b = 2.97$, 95% CI [2.72, 3.21], $F(1, 82.0) = 587.87$; fear, $b = 3.47$, 95% CI [3.23, 3.72], $F(1, 82.0) = 801.23$; sadness, $b = 3.06$, 95% CI [2.83, 3.30], $F(1, 82.1) = 697.22$; joy, $b = 3.18$, 95% CI [2.88, 3.47], $F(1, 82.1) = 468.07$; all $p_{FWE} < .001$.

Contrary to predictions, PCL-R Total scores were unrelated to intensity ratings of the prototypical emotion expressions (collapsed across categories) relative to neutral trials, $b = -0.02$, 95% CI [-0.03, 0.01], $F(1, 80.1) = 1.86$, $p = .176$. Similarly, PCL-R Total scores were unrelated to intensity ratings for each emotion category relative to neutral trials: anger, $b = -0.02$, 95% CI [-0.05, 0.02], $F(1, 80.0) = 0.97$; disgust, $b = -0.02$, 95% CI [-0.05, 0.01], $F(1, 80.1) = 1.56$; fear, $b = -0.02$, 95% CI [-0.05, 0.01], $F(1, 80.0) = 1.57$; sadness, $b = -0.02$, 95% CI [-0.05, 0.01], $F(1, 80.1) = 1.58$; joy, $b = -0.02$, 95% CI [-0.06, 0.02], $F(1, 80.1) = 1.22$; all $p_{FWE} > .325$.

PCL-R Factor 1 scores were also unrelated to intensity ratings of prototypical emotion expressions (collapsed across categories) relative to neutral trials, $b = -0.08$, 95% CI [-0.18, 0.02], $F(1, 77.9) = 2.60$, $p = .111$, and of specific emotion categories relative to neutral trials: anger, $b = -0.04$, 95% CI [-0.16, 0.07], $F(1, 79.7) = 0.57$; disgust, $b = -0.07$, 95% CI [-0.18, 0.04], $F(1, 79.6) = 1.60$; fear, $b = -0.07$, 95% CI [-0.18, 0.04], $F(1, 80.0) = 1.42$; sadness, $b = -0.10$, 95% CI [-0.20, 0.00], $F(1, 79.7) = 3.65$; and joy, $b = -0.12$, 95% CI [-0.25, 0.02], $F(1, 79.8) = 3.11$; all $p_{FWE} > .299$. PCL-R Factor 2 scores were similarly unrelated to intensity ratings of prototypical emotion expressions (collapsed across categories) relative to neutral trials, $b = 0.02$, 95% CI [-0.05, 0.08], $F(1, 77.6) = 0.32$, $p = .572$, and of specific emotion categories relative to neutral trials: anger, $b = 0.00$, 95% CI [-0.07, 0.07], $F(1, 79.7) = 0.00$; disgust, $b = 0.01$, 95% CI [-0.06, 0.08], $F(1, 79.3) = 0.11$; fear, $b = 0.01$, 95% CI [-0.06, 0.08], $F(1, 79.9) = 0.11$; sadness, $b = 0.04$, 95% CI [-0.03, 0.10], $F(1, 79.1) = 1.04$; and joy, $b = 0.03$, 95% CI [-0.05, 0.12], $F(1, 79.5) = 0.52$; all $p_{FWE} > .964$.

## Facial mimicry

Across participants, EMG z-scores failed to demonstrate facial mimicry of prototypical emotion expressions (collapsed across categories) relative to neutral trials (Table 3): *corrugator*, $b =$

**Table 3. Physiological responses across participants ($N$ = 88).**

| Physiological Measure | | All Emotion Categories† | Emotion Categories | | | | | Neutral |
|---|---|---|---|---|---|---|---|---|
| | | | Anger | Disgust | Fear | Sadness | Joy | |
| *Corrugator* | M | -0.11 | -0.03 | 0.00 | -0.09 | -0.04 | -0.41 | -0.06 |
| | SD | 0.33 | 0.36 | 0.53 | 0.68 | 0.49 | 0.45 | 0.28 |
| *Levator* | M | -0.03 | -0.04 | 0.13 | -0.08 | -0.10 | -0.06 | -0.03 |
| | SD | 0.23 | 0.22 | 0.57 | 0.31 | 0.17 | 0.30 | 0.23 |
| *Zygomaticus* | M | -0.03 | -0.03 | 0.03 | -0.04 | -0.14 | 0.02 | -0.03 |
| | SD | 0.25 | 0.28 | 0.26 | 0.34 | 0.28 | 0.53 | 0.22 |
| *Depressor* ($n$ = 66) | M | -0.19 | -0.29 | -0.10 | -0.16 | -0.19 | -0.19 | -0.12 |
| | SD | 0.37 | 0.62 | 0.42 | 0.44 | 0.55 | 0.62 | 0.65 |
| SCR ($n$ = 87) | M | 0.26 | 0.26 | 0.25 | 0.26 | 0.25 | 0.26 | 0.25 |
| | SD | 0.19 | 0.22 | 0.18 | 0.22 | 0.20 | 0.21 | 0.20 |
| Cardiac Deceleration ($n$ = 86) | M | -2.69 | -2.65 | -2.60 | -2.34 | -3.14 | -2.72 | -2.93 |
| | SD | 1.76 | 2.21 | 2.22 | 1.84 | 2.45 | 2.14 | 2.30 |

Note.

† Responses were collapsed across emotion categories except neutral.

-0.05, 95% CI [-0.14, 0.03], $F(1, 82.1)$ = 1.64; *levator*, $b$ = 0.01, 95% CI [-0.06, 0.07], $F(1, 82.0)$ = 0.05; *zygomaticus*, $b$ = 0.00, 95% CI [-0.07, 0.06], $F(1, 81.9)$ = 0.01; *depressor* $b$ = -0.03, 95% CI [-0.21, 0.16], $F(1, 47.8)$ = 0.07; all $p$ > .204. Similarly, EMG z-scores failed to demonstrate facial mimicry of each emotion category relative to neutral trials: anger *corrugator*, $b$ = 0.03, 95% CI [-0.06, 0.11], $F(1, 82.1)$ = 0.43; disgust *levator*, $b$ = 0.17, 95% CI [0.04, 0.29], $F(1, 82.0)$ = 6.69; fear *corrugator*, $b$ = -0.03, 95% CI [-0.18, 0.12], $F(1, 82.0)$ = 0.14; fear *zygomaticus*, $b$ = -0.01, 95% CI [-0.09, 0.07], $F(1, 81.9)$ = 0.10; sadness *depressor*, $b$ = -0.08, 95% CI [-0.26, 0.11], $F(1, 73.9)$ = 0.69; joy *zygomaticus*, $b$ = 0.04, 95% CI [-0.07, 0.15], $F(1, 81.8)$ = 0.61; all $p_{FWE}$ > .065. As a follow-up analysis, we examined EMG z-scores collapsed across prototypical emotion expressions but not contrasted with neutral trials. Unexpectedly, participants displayed less *depressor* activity in response to prototypical sadness expressions, $b$ = -0.20, 95% CI [-0.32, -0.06], $F(1, 49.7)$ = 9.52, $p_{FWE}$ = .018. Participants did not display significant facial muscle response to the other prototypical emotion expressions: anger *corrugator*, $b$ = -0.04, 95% CI [-0.11, 0.04], $F(1, 83.0)$ = 0.86; disgust *levator*, $b$ = 0.14, 95% CI [0.01, 0.26], $F(1, 82.6)$ = 4.98; fear *corrugator*, $b$ = -0.09, 95% CI [-0.23, 0.06], $F(1, 82.7)$ = 1.45; fear *zygomaticus*, $b$ = -0.04, 95% CI [-0.11, 0.03], $F(1, 82.9)$ = 1.07; joy *zygomaticus*, $b$ = 0.03, 95% CI [-0.08, 0.14], $F(1, 80.0)$ = 0.34; all $p_{FWE}$ > .139.

Contrary to predictions, PCL-R Total scores were unrelated to mimicry of prototypical emotion expressions (collapsed across categories) relative to neutral trials: *corrugator*, $b$ = 0.00, 95% CI [-0.01, 0.01], $F(1, 80.0)$ = 0.07; *levator*, $b$ = 0.00, 95% CI [-0.01, 0.01], $F(1, 79.8)$ = 0.46; *zygomaticus*, $b$ = 0.00, 95% CI [-0.01, 0.00], $F(1, 79.7)$ = 0.81; *depressor*, $b$ = 0.01, 95% CI [-0.01, 0.03], $F(1, 72.1)$ = 0.71; all $p$ > .370. PCL-R Total scores were also unrelated to mimicry of specific emotion categories relative to neutral trials: anger *corrugator*, $b$ = 0.00, 95% CI [-0.01, 0.02], $F(1, 80.0)$ = 0.59; disgust *levator*, $b$ = -0.01, 95% CI [-0.02, 0.01], $F(1, 80.0)$ = 0.46; fear *corrugator*, $b$ = -0.01, 95% CI [-0.03, 0.01], $F(1, 80.0)$ = 0.43; fear *zygomaticus*, $b$ = 0.00, 95% CI [-0.01, 0.01], $F(1, 80.0)$ = 0.01; sadness *depressor*, $b$ = 0.01, 95% CI [-0.01, 0.04], $F(1, 76.3)$ = 1.50; and joy *zygomaticus*, $b$ = -0.01, 95% CI [-0.02, 0.01], $F(1, 79.9)$ = 1.47; all $p_{FWE}$ > .931.

PCL-R Factor 1 scores were also unrelated to facial mimicry of prototypical emotion expressions (collapsed across categories) relative to neutral trials: *corrugator*, $b$ = 0.01, 95% CI

[-0.03, 0.05], $F(1, 78.0) = 0.35$, $p = .557$; *levator*, $b = 0.00$, 95% CI [-0.03, 0.03], $F(1, 77.9) = 0.05$, $p = .829$; *zygomaticus*, $b = 0.01$, 95% CI [-0.01, 0.04], $F(1, 77.8) = 0.90$, $p = .345$; *depressor*, $b = 0.00$, 95% CI [-0.07, 0.07], $F(1, 67.2) = 0.00$, $p = 996$. Nor were PCL-R Factor 1 scores related to facial mimicry of specific emotion categories relative to neutral trials: anger *corrugator*, $b = 0.03$, 95% CI [-0.01, 0.06], $F(1, 78.0) = 2.07$; disgust *levator*, $b = -0.02$, 95% CI [-0.08, 0.04], $F(1, 77.8) = 0.32$; fear *corrugator*, $b = 0.00$, 95% CI [-0.07, 0.07], $F(1, 78.0) = 0.01$; fear *zygomaticus*, $b = 0.03$, 95% CI [-0.01, 0.06], $F(1, 77.8) = 1.93$; sadness *depressor*, $b = 0.00$, 95% CI [-0.08, 0.08], $F(1, 75.7) = 0.01$; and joy *zygomaticus*, $b = -0.02$, 95% CI [-0.07, 0.03], $F(1, 78.0) = 0.43$; all $p_{FWE} > .844$. PCL-R Factor 2 scores were similarly unrelated to facial mimicry of prototypical emotion expressions (collapsed across categories) relative to neutral trials: *corrugator*, $b = 0.00$, 95% CI [-0.03, 0.02], $F(1, 77.8) = 0.06$, $p = .812$; *levator*, $b = 0.00$, 95% CI [-0.02, 0.02], $F(1, 77.8) = 0.09$, $p = .765$; *zygomaticus*, $b = -0.01$, 95% CI [-0.03, 0.00], $F(1, 77.7) = 2.45$, $p = .122$; *depressor*, $b = 0.02$, 95% CI [-0.03, 0.07], $F(1, 71.3) = 0.73$, $p = .395$. Nor were PCL-R Factor 2 scores related to facial mimicry of specific emotion categories relative to neutral trials: anger *corrugator*, $b = -0.01$, 95% CI [-0.03, 0.02], $F(1, 77.9) = 0.43$; disgust *levator*, $b = 0.00$, 95% CI [-0.04, 0.04], $F(1, 77.8) = 0.02$; fear *corrugator*, $b = -0.01$, 95% CI [-0.06, 0.03], $F(1, 78.0) = 0.33$; fear *zygomaticus*, $b = -0.02$, 95% CI [-0.04, 0.01], $F(1, 77.9) = 1.64$; sadness *depressor*, $b = 0.03$, 95% CI [-0.02, 0.08], $F(1, 77.1) = 1.08$; and joy *zygomaticus*, $b = 0.00$, 95% CI [-0.04, 0.03], $F(1, 77.8) = 0.07$; all $p_{FWE} > .879$.

## Skin conductance response

Participants did not display greater SCR amplitude in response to prototypical emotion expressions (collapsed across categories) relative to neutral trials (Table 3), $b = 0.02$, 95% CI [-0.01, 0.04], $F(1, 69.4) = 1.21$, $p = .276$. Similarly, SCR amplitude did not differ between each emotion category and neutral trials: anger, $b = 0.01$, 95% CI [-0.02, 0.05], $F(1, 74.0) = 0.40$; disgust, $b = 0.00$, 95% CI [-0.03, 0.03], $F(1, 77.4) = 0.05$; fear, $b = 0.01$, 95% CI [-0.02, 0.04], $F(1, 76.7) = 0.20$; sadness, $b = 0.00$, 95% CI [-0.03, 0.03], $F(1, 76.0) = 0.01$; joy, $b = 0.01$, 95% CI [-0.02, 0.04], $F(1, 76.6) = 0.35$; all $p_{FWE} > .936$. As a follow-up analysis, we examined SCR amplitude collapsed across prototypical emotion expressions but not contrasted with neutral trials. As expected, participants displayed significant SCR to the prototypical emotion expressions excluding neutral trials, $b = 0.27$, 95% CI [0.23, 0.31], $F(1, 82.0) = 175.91$, $p < .001$.

Contrary to predictions, PCL-R Total scores were unrelated to SCR amplitude in response to prototypical emotion expressions (collapsed across categories) relative to neutral trials, $b = 0.00$, 95% CI [0.00, 0.00], $F(1, 69.7) = 0.25$, $p = .616$. PCL-R Total scores were unrelated to SCR amplitude for each emotion category relative to neutral trials: anger, $b = 0.00$, 95% CI [0.00, 0.00], $F(1, 71.1) = 0.00$; disgust, $b = 0.00$, 95% CI [0.00, 0.01], $F(1, 75.9) = 0.98$; fear, $b = 0.00$, 95% CI [0.00, 0.01], $F(1, 72.0) = 0.24$; sadness, $b = 0.00$, 95% CI [0.00, 0.00], $F(1, 73.9) = 0.09$; joy, $b = 0.00$, 95% CI [0.00, 0.01], $F(1, 74.6) = 0.36$; all $p_{FWE} > .952$.

PCL-R Factor 1 scores were also unrelated to SCR amplitude in response to prototypical emotion expressions (collapsed across categories) relative to neutral trials, $b = 0.00$, 95% CI [-0.01, 0.01], $F(1, 76.0) = 0.00$, $p = .947$, and for specific emotion categories relative to neutral trials: anger, $b = 0.00$, 95% CI [-0.02, 0.02], $F(1, 74.3) = 0.00$; disgust, $b = 0.01$, 95% CI [-0.01, 0.02], $F(1, 75.7) = 0.66$; fear, $b = 0.00$, 95% CI [-0.01, 0.01], $F(1, 77.0) = 0.00$; sadness, $b = 0.00$, 95% CI [-0.02, 0.01], $F(1, 75.2) = 0.10$; and joy, $b = 0.00$, 95% CI [-0.01, 0.02], $F(1, 75.6) = 0.03$; all $p_{FWE} > .990$. PCL-R Factor 2 scores were similarly unrelated to SCR amplitude in response to prototypical emotion expressions (collapsed across categories) relative to neutral trials, $b = 0.00$, 95% CI [-0.01, 0.01], $F(1, 68.5) = 0.13$, $p = .719$, and for specific emotion categories relative to neutral trials: anger, $b = 0.00$, 95% CI [-0.01, 0.01], $F(1, 60.9) = 0.03$; disgust,

$b = 0.00$, 95% CI [-0.01, 0.01], $F(1, 67.8) = 0.01$; fear, $b = 0.00$, 95% CI [-0.01, 0.01], $F(1, 72.9) = 0.14$; sadness, $b = 0.00$, 95% CI [-0.01, 0.01], $F(1, 68.6) = 0.27$; and joy, $b = 0.00$, 95% CI [-0.01, 0.01], $F(1, 66.1) = 0.28$; all $p_{FWE} > .917$.

## Cardiac deceleration

Participants did not display greater cardiac deceleration to prototypical emotion expressions collapsed across categories relative to neutral trials (Table 3), $b = 0.23$, 95% CI [-0.10, 0.56], $F(1, 71.7) = 1.92$, $p = .170$. However, participants showed significantly diminished cardiac deceleration in response to prototypical expressions of fear than to neutral trials, $b = 0.57$, 95% CI [0.17, 0.95], $F(1, 73.8) = 8.23$, $p_{FWE} = .025$. Cardiac deceleration did not differ between the other emotion categories and neutral trials: anger, $b = 0.30$, 95% CI [-0.12, 0.71], $F(1, 77.2) = 2.04$; disgust, $b = 0.31$, 95% CI [-0.11, 0.73], $F(1, 74.6) = 2.11$; sadness, $b = -0.20$, 95% CI [-0.66, 0.27], $F(1, 76.2) = 0.69$; joy, $b = 0.22$, 95% CI [-0.21, 0.64], $F(1, 80.0) = 1.00$; all $p_{FWE} > .407$. As a follow-up analysis, we examined cardiac deceleration collapsed across prototypical emotion expressions but not contrasted with neutral trials. As expected, participants displayed significant cardiac deceleration to the prototypical emotion expressions excluding neutral trials, $b = -2.67$, 95% CI [-3.01, -2.33], $F(1, 80.8) = 244.83$, $p < .001$.

Contrary to predictions, PCL-R Total scores were unrelated to cardiac deceleration in response to prototypical emotion expressions (collapsed across categories) relative to neutral trials, $b = -0.04$, 95% CI [-0.08, 0.01], $F(1, 72.9) = 3.08$, $p = .083$. Further, PCL-R Total scores were unrelated to cardiac deceleration for any emotion category relative to neutral trials: anger, $b = -0.05$, 95% CI [-0.10, 0.00], $F(1, 76.5) = 3.46$; disgust, $b = -0.04$, 95% CI [-0.10, 0.01], $F(1, 70.6) = 2.17$; fear, $b = -0.03$, 95% CI [-0.08, 0.02], $F(1, 78.0) = 1.17$; sadness, $b = -0.06$, 95% CI [-0.11, 0.00], $F(1, 78.8) = 3.61$; joy, $b = -0.01$, 95% CI [-0.06, 0.05], $F(1, 77.2) = 0.03$; all $p_{FWE} > .267$.

PCL-R Factor 1 scores were also unrelated to cardiac deceleration in response to prototypical emotion expressions (collapsed across categories) relative to neutral trials, $b = -0.02$, 95% CI [-0.17, 0.12], $F(1, 79.5) = 0.10$, $p = .750$, and for specific emotion categories relative to neutral trials: anger, $b = -0.03$, 95% CI [-0.10, 0.29], $F(1, 78.3) = 0.11$; disgust, $b = -0.11$, 95% CI [-0.29, 0.08], $F(1, 79.9) = 1.32$; fear, $b = -0.13$, 95% CI [-0.30, 0.04], $F(1, 79.9) = 2.13$; sadness, $b = 0.07$, 95% CI [-0.14, 0.27], $F(1, 79.0) = 0.42$; and joy, $b = 0.09$, 95% CI [-0.10, 0.29], $F(1, 79.8) = 0.90$; all $p_{FWE} > .664$. PCL-R Factor 2 scores were similarly unrelated to cardiac deceleration in response to prototypical emotion expressions collapsed across categories relative to neutral trials, $b = -0.05$, 95% CI [-0.14, 0.05], $F(1, 75.6) = 0.98$, $p = .325$, and for specific emotion categories relative to neutral trials: anger, $b = -0.06$, 95% CI [-0.19, 0.07], $F(1, 74.5) = 1.04$; disgust, $b = 0.00$, 95% CI [-0.12, 0.12], $F(1, 74.6) = 0.00$; fear, $b = 0.03$, 95% CI [-0.08, 0.14], $F(1, 77.6) = 0.26$; sadness, $b = -0.13$, 95% CI [-0.26, 0.01], $F(1, 75.7) = 3.44$; and joy, $b = -0.06$, 95% CI [-0.19, .07], $F(1, 75.4) = 0.74$; all $p_{FWE} > .339$.

## Discussion

This study sought to identify dysfunctional physiological responses to dynamic prototypical facial emotion expressions in psychopathy. Psychopathy was predicted to be related to reduced emotion categorization accuracy, ratings of valence and intensity, facial mimicry and autonomic arousal to prototypical emotion expressions. We hypothesized alterations in relation to the total construct of psychopathy and to the interpersonal/affective features of psychopathy, in particular. The data supported none of these hypotheses. Overall, participants categorized facial emotion expressions more accurately than chance and rated the valence and intensity of the target person's emotion appropriately. Contrary to our hypotheses, psychopathy was

unrelated to emotion categorization accuracy, valence ratings, and intensity ratings. The prototypical facial expression task failed to elicit facial mimicry from the full sample of incarcerated men, which does not allow drawing conclusions about the relationship between psychopathy and facial mimicry. Although participants did not display greater autonomic arousal in response to prototypical emotion expressions relative to neutral trials, they displayed autonomic arousal when neutral trials were excluded from the model. This suggests that the facial stimuli, but not the prototypical emotion expressions per se, elicited autonomic arousal. Psychopathy was unrelated to cardiac deceleration or SCR amplitude. For the sake of transparency, we note that we modified two analytic choices (as described in the Methods section) based on reviewer recommendations. Both our original analyses and the current, reviewer-recommended analyses tested the same hypotheses and yielded similarly null results.

Contrary to two earlier meta-analyses [2,3], in the current study psychopathy was unrelated to emotion categorization accuracy. These meta-analyses found that people high in psychopathy categorize prototypical facial emotion expressions with diminished accuracy. The previous meta-analytic findings were based on data collapsed across emotion categories and when prototypical expressions of fear, sadness, happiness, and surprise were examined separately. We have considered the following explanations for the present null findings. First, a ceiling effect in the current study (90% categorization accuracy for all emotion categories across participants) may possibly have prevented the observation of a deficit in relation to psychopathy. Second, omitting prototypical surprise expressions from the present study's stimuli might have reduced the observed relationship between psychopathy and emotion categorization accuracy. However, this explanation seems unlikely, given that meta-analyses found categorization deficits for emotion categories that were included in the present study's stimuli, including fear, sadness, and happiness [2,3]. Third, the origin of the current sample (an incarcerated population) may have influenced the null results. Once again, this explanation seems unlikely, given that the origin of the sample (incarcerated vs. community) does not appear to influence the relationship between psychopathy and emotion categorization accuracy [3]. Therefore, the current study joins other studies that have found no facial emotion categorization deficit in individuals high in psychopathy [4,60,61].

Some researchers have contested the idea that even healthy individuals can "read" others' emotions from facial muscle configurations, or that prototypical facial emotion expressions reliably correspond to a given emotional state [31,32]. People's perceptions of emotion based on facial configuration depend on context [62–64] and vary across cultures [65]. Additionally, facial expressions appear to convey information about a person's experience of valence (i.e., pleasantness) and arousal [i.e., energetic activation; 31,34]. Thus, participants also rated the valence and intensity (but not arousal) of the other person's emotion. Psychopathy was unrelated to valence or intensity ratings of the other person's emotion. Only one previous study employed a similar method and observed that women high in psychopathy judged prototypical joy faces to be less positive, but observed no differences in valence ratings for other emotion categories [37]. This previous study also found that women high in psychopathy rated a variety of prototypical facial emotion expressions as less emotionally arousing to themselves. Notably, neither the previous study nor the current study gathered ratings of the other person's state of arousal (ranging from energized to calm). Further research could help to clarify whether individuals high in psychopathy judge facial expressions to be more neutral in terms of valence and arousal.

Unexpectedly, the stimuli failed to elicit facial mimicry from participants, even those low in psychopathy. The null relationship between psychopathy and facial mimicry should therefore be interpreted with this caveat. This null relationship across participants was unexpected, given that we replicated the EMG data analysis steps of a prior study that observed

spontaneous facial mimicry in a non-incarcerated sample [66]. We elaborate on the missing facial mimicry in the full sample in the limitations paragraph at the end of this section. Only two prior studies have examined the relationship between psychopathic traits and spontaneous mimicry of dynamic prototypical facial emotion expressions. One found evidence of reduced *zygomaticus* mimicry of prototypical joy expressions and reduced *corrugator* mimicry of prototypical anger and sadness expressions among juveniles with callous-unemotional traits [12]. However, a study of incarcerated adult men found no relationship between psychopathy and *corrugator* mimicry of prototypical anger and sadness expressions [13]. Interestingly, psychopathy in adulthood has been related to reduced contagious yawning [67,68], another process of reproducing another's bodily state that may be related to facial mimicry [69]. Related studies examining the ability to produce prototypical facial expressions have yielded mixed evidence for abnormalities in facial muscle activity in psychopathy. These studies have found that psychopathy is related to increased use of typical muscles [70], decreased production of appropriate facial expressions to negative static pictures [71], impairments in deliberate mimicry and production of facial expressions that were attributable to deficits in general mental ability [72], and no deficit making or inhibiting appropriate facial expressions [73]. Thus, the few existing studies have found inconsistent evidence that the ability to mimic or produce prototypical facial emotion expressions is impaired in psychopathy. Future directions in this field should therefore investigate potential moderators, such as motivational factors [74].

The data failed to provide support for the hypothesis that psychopathy is related to reduced sympathetic arousal to facial expressions. Psychopathy has previously been linked to reduced SCR during tasks, especially to negative stimuli [75]. To our knowledge, this is the first study of psychopathy to derive SCR amplitude using dynamic causal modeling, a method that allows for inference about sympathetic arousal, rather than just skin conductance [76]. This method has been shown to outperform peak detection methods (which were used by prior studies of psychopathy) when predicting healthy individuals' sympathetic arousal to emotional stimuli [77]. The divergence in SCR findings (which were not significant in the current study and significant in prior studies of psychopathy) could possibly have resulted from differences in SCR amplitude estimation (dynamic causal modeling in the current study and peak detection in prior studies). Dynamic causal modeling would allow future studies of psychopathy to draw more direct inferences about sympathetic arousal to other socioemotional stimuli. Alternatively, heterogeneity among individuals high in psychopathy might help to explain the present null findings regarding sympathetic arousal. Future studies might examine whether two subtypes of psychopathy, distinguished by low versus high levels of trait negative affect and anxiety [78–82], display different patterns of sympathetic arousal to socioemotional stimuli such as faces. Though the current sample of individuals high in psychopathy (PCL-R $\geq$ 30) was too small to test the hypothesis ($n$ = 30), prior work has reported reduced sympathetic arousal to socioemotional stimuli specifically among individuals high in psychopathy with low levels of negative affect and anxiety [23,24].

Several limitations need to be considered for the interpretation of the current data. The lack of facial mimicry exhibited by the current sample presents a clear limitation to drawing conclusions about mimicry in psychopathy. It is possible that facial mimicry is less pronounced among incarcerated samples than community samples. In general, incarcerated samples differ from community samples along dimensions that may affect mimicry in response to white faces, including having higher prevalence of mental illness [83] and traumatic brain injury [84], greater proportion of racial and ethnic minority groups [85], and lower socioeconomic status [86]. However, Künecke and colleagues found no facial mimicry differences between incarcerated and non-incarcerated individuals [13]. Future studies might alter the methods of the current study to measure facial mimicry more sensitively. For example, the current study

measured participants' facial muscle activity via EMG because it is a perceiver-independent measure of muscle contractions that may not be visible to the naked eye [87]. However, the four sets of electrodes attached to participants' faces may have obtruded spontaneous mimicry; most facial mimicry studies have used only one or two sets of electrodes [7,88]. Although, we can note that prior studies have observed facial mimicry with four sets of electrodes [66]. Future studies might use lightweight printed electrodes to reduce the likelihood of obtruding spontaneous facial muscle activity [89]. The EMG impedance threshold, which was higher than some published guidelines [90], is another methodological limitation of the study. Although we were able to detect deliberate facial muscle movements (see S1 File), high EMG impedance could have prevented detection of more subtle muscle movements. Furthermore, though dynamic stimuli elicit more mimicry than static stimuli [11,91], the videos of posed, prototypical expressions may have engendered less mimicry than genuine expressions would have. This explanation seems unlikely, however, given previous studies' findings of comparable facial mimicry of genuine and posed facial expressions [7,92]. Future studies of facial mimicry in psychopathy may benefit from recruiting non-incarcerated samples, which consistently display spontaneous mimicry [7,66,91].

Although the measurement of SCR and cardiac deceleration allows for the estimation of activity in the sympathetic nervous system and parasympathetic nervous system, respectively [77,93], more direct measures of sympathetic and parasympathetic activity could be used in future studies. Heart rate, in particular, from which cardiac deceleration was calculated, is affected by both sympathetic and parasympathetic activity. Additionally, heart rate measured from the finger by pulse plethysmography is also influenced by artery diameter [94]. Thus, the observed cardiac deceleration in the current study may have been influenced by a variety of factors. Future studies might derive purer estimations of sympathetic (e.g., pre-ejection period) and parasympathetic activity (e.g., respiratory sinus arrhythmia) via electrocardiography, impedance cardiography, and respiration belt [95]. Research examining these measures of autonomic activity in adults with psychopathy as they respond to socioemotional stimuli is currently lacking [although see 96].

## Conclusions

The current study constituted a novel attempt to identify disrupted physiological mechanisms contributing to impaired processing of others' prototypical dynamic facial expressions of emotion associated with psychopathy. The results failed to identify aberrant behavioral and physiological responses to prototypical facial emotion expressions in psychopathy.

## Supporting information

**S1 Fig. Average time series for the four deliberate muscle movements across participants.**
(TIF)

**S1 Table. Zero order correlations among continuous independent variables and dependent variables.**
(DOCX)

**S1 File. Supplemental analyses.**
(DOCX)

## Acknowledgments

We thank the many individuals at the Wisconsin Department of Corrections who made this research possible, and are especially indebted to Deputy Warden Tom Nickel, Warden Randy Hepp, and Dr. Kevin Kallas. We also thank Andrew Langbehn, Louis Monette, Nicole Huth, Erica Gelman, and Mateo Vargas-Nunez for their efforts toward this study.

## Author Contributions

**Conceptualization:** Philip Deming, Hedwig Eisenbarth, Michael Koenigs.

**Formal analysis:** Philip Deming, Odile Rodrik, Shelby S. Weaver.

**Investigation:** Philip Deming.

**Methodology:** Philip Deming, Hedwig Eisenbarth, Michael Koenigs.

**Project administration:** Philip Deming.

**Resources:** Kent A. Kiehl, Michael Koenigs.

**Supervision:** Michael Koenigs.

**Visualization:** Philip Deming.

**Writing – original draft:** Philip Deming.

**Writing – review & editing:** Philip Deming, Hedwig Eisenbarth, Michael Koenigs.

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
