## [Decision Letter · Decision Letter 0]

23 Feb 2022

PONE-D-21-38079An examination of autonomic and facial responses to prototypical facial emotion expressions in psychopathyPLOS ONE

Dear Dr. Deming,

Thank you for submitting your manuscript to PLOS ONE. After careful consideration, we feel that it has some merit but does not fully meet PLOS ONE’s publication criteria as it currently stands. Therefore, we invite you to submit a revised version of the manuscript that addresses the points raised during the review process.

ACADEMIC EDITOR:

I was able to obtain two reviews from colleagues knowledgeable in the areas of psychopathy, emotion, and/or psychophysiology. I reviewed the paper independent of the reviewers. There was broad agreement that the paper had clear themes and focus, and that the experimental methods were fairly rigorous. There was evidence of transparency in the reporting of methods and results, although as noted by Reviewer 1, pre-registration was not included. The reviewers and I noted several limitations regarding the analyses, results, and interpretation of the findings that may be addressed in a greatly revised version of the paper. This is no guarantee of publication though, given the substantial weaknesses noted.

The physiological data reduction choices need more justification. Different baseline windows were used across EMG and cardiac deceleration, and it is unclear why that is. Reviewer 1 made relevant comments in this regard and provided several other suggestions for you to seriously consider.

I’m having trouble understanding the rationale for not comparing the emotional faces to neutral faces when analyzing valence and intensity rating, as well as physio responses. These comparisons account for between subject variance in general arousal or response tendencies, to get at unique effects of emotional stimuli on response modalities. In some cases in your paper, only descriptive statistics are reported to conclude whether there was sufficient activity or evidence of mimicry. More common analyses involve testing within-subject effects of face category (with specific contrasts of the emotional vs. neutral facets) to examine basic effects, and then including psychopathy scores as between subject factor moderating effects of face category. The absence of these comparisons clouds the potential contribution or meaning of the data.

Reviewer 2 makes important points about the reporting of results and ambiguousness in interpretation. I agree that in general, if you are using NHST, you should stick to the p values you determine. The interpretation of trend level p values is problematic, and when you report results both with and without outliers, it is unclear which results we should rely on and what is the rationale for the choice. What were the planned analyses in terms of outlier and covariate inclusion criteria? This can help determine which results are primary in your interpretation. Please also include effect size estimates to understand the impact and potential meaningfulness of the results. If effects are small, indicate whether this size of effects can still provide important information and how.

Neither reviewer (and I agree) found the exploratory analyses particularly compelling or meaningful, especially due to the inconsistent manner of the groupings and small sample sizes. Instead, it would have been more straightforward and consistent with theory to examine factor or facet level relationships with mimicry and autonomic arousal to emotion faces. The primary focus of the paper on total psychopathy, despite persistent evidence of differential relationships between specific psychopathic traits and emotional functioning is confusing. As noted by Reviewer 2, there are choices in hypotheses and analyses that are not consistent with prior work or theory. This should be noted explicitly.

We look forward to receiving your revised manuscript.

Kind regards,

Edelyn Verona

Academic Editor

PLOS ONE

Journal Requirements:

2.  Please provide additional information regarding the considerations made for the prisoners included in this study. For instance, please discuss whether participants were able to opt out of the study and whether individuals who did not participate receive the same treatment offered to participants.

3. We note that you have stated that you will provide repository information for your data at acceptance. Should your manuscript be accepted for publication, we will hold it until you provide the relevant accession numbers or DOIs necessary to access your data. If you wish to make changes to your Data Availability statement, please describe these changes in your cover letter and we will update your Data Availability statement to reflect the information you provide

4. We note that your paper includes detailed descriptions of individual patients/participants. As per the PLOS ONE policy (http://journals.plos.org/plosone/s/submission-guidelines#loc-human-subjects-research) on papers that include identifying, or potentially identifying, information, the individual(s) or parent(s)/guardian(s) must be informed of the terms of the PLOS open-access (CC-BY) license and provide specific permission for publication of these details under the terms of this license. Please download the Consent Form for Publication in a PLOS Journal (http://journals.plos.org/plosone/s/file?id=8ce6/plos-consent-form-english.pdf). The signed consent form should not be submitted with the manuscript, but should be securely filed in the individual's case notes. Please amend the methods section and ethics statement of the manuscript to explicitly state that the patient/participant has provided consent for publication: “The individual in this manuscript has given written informed consent (as outlined in PLOS consent form) to publish these case details

Reviewers' comments:

Reviewer's Responses to Questions

**Comments to the Author**

1. Is the manuscript technically sound, and do the data support the conclusions?

Reviewer #1: Partly

Reviewer #2: Partly

2. Has the statistical analysis been performed appropriately and rigorously? 

Reviewer #1: No

Reviewer #2: I Don't Know

3. Have the authors made all data underlying the findings in their manuscript fully available?

Reviewer #1: No

Reviewer #2: Yes

4. Is the manuscript presented in an intelligible fashion and written in standard English?

Reviewer #1: Yes

Reviewer #2: Yes

5. Review Comments to the Author

Reviewer #1: This intriguing study provides a multimodal test of the impact of facial expressions on those with psychopathic traits, going beyond mere facial mimicry to examine autonomic functioning in addition. However, the critical subgrouping did not proceed in the way the authors analyzed their data, making it challenging to understand the contribution of this work to the psychopathy literature. Further aspects of the data reduction bear scrutiny to understand the panoply of surprising null results.

INTRODUCTION

The introduction does a nice job of setting up the rationale for the study. However, I couldn’t find the link to the registration that would allow me to verify which analyses were confirmatory and which were exploratory. What is the link to that registration?

Also, to what extent are PCL-R factors differentially associated with facial recognition deficits?

METHOD

Why were seven participants missing Welsh anxiety data? Were the profiles uninterpretable for some reason? If so, the specific validity scales for those exclusions should be given. Or were the data just missing? In either case, if the MPQ data were valid for those participants, some kind of imputation procedure should be used to impute these scores. Otherwise, the participants analyzed in the covariate analyses are not the same as those in the whole sample (eliminating almost 10% of the whole-sample participants), which distorts the meaning of the covariate analyses relative to those in the whole sample.

What were the PEM, NEM, and CON scores on the MPQ to be placed in Table 1? T scores would be preferable to report for such analyses to make the interpretation easier, as in the IQ measure.

The term “Caucasian” should be deprecated in referring to White people, as that particular term has a long (and long-criticized) history implying racial superiority of individuals so labeled: https://www.ncbi.nlm.nih.gov/pmc/articles/PMC1444385/pdf/bmjcred00490-0048.pdf

A 25 kΩ impedance is relatively high compared to published guidelines; why was such a high level of impedance considered acceptable in this study?

Why was a 60 Hz notch filter used in this dataset? Was there evidence of line noise? Using a 60 Hz filter without line noise can introduce noise and ripples into the data that would not otherwise exist.

If the facial stimulus started changing 1 s after the face’s onset and reached peak expression at 3 s – 4 s, why was not the EMG scoring window from 2 s – 5 s to account for the cited 1 s delay in EMG onset once an expression is evident and the stimulus parameters described above? Or was this procedure meant just to examine psychopathy’s relationship to any facial stimulus (given the ramp-up in the supplementary figure) rather than mimicry specifically?

The SCR data analysis was clever. I assume the canonical function was estimated for the entire 6 s of stimulus presentation? How long was the ratings presentation?

For the cardiac deceleration data, would it not make sense to have a similar analytical window as the EMG data proposed above? That way, decelerations not due to a particular facial expression would be excluded. Or were the researchers interested only in the deceleration to human faces, irrespective of their portrayed emotional expression?

I liked the authors’ justification for choosing only target muscles for investigating particular target emotions, as those analytical decisions follow well from the theoretical literature. In-text citations would help defend those even more strongly outside of the supplementary materials.

Were covariates centered in the analyses involving them? Not doing so can eliminate main effects of interest.

From the authors’ exploratory analyses described in the introduction, it seems that the Welsh anxiety x PCL-R total score interaction should have been entered – or was it, and I just misunderstood the terms in the statistical model?

The clustering approach described in the supplementary analyses does not identify specific subtypes of individuals with psychopathy. Instead, it will divide the overall sample into two groups across all levels of the PCL-R.. Also, it appears that two participants were excluded from the high psychopathy group (n = 30) when the subtype analyses were conducted (n = 28); why were they missing? I assume these were the two participants for whom MPQ data were not available.

Finally, k means cluster analyses do not have strong stopping rules; model-based cluster analyses would be preferable to examine whether two groups or more are optimal in these data with the ability to quantify the probability of each participant’s membership in each cluster.

It may be preferable to describe the subtype variables as reference-coded coded rather than dummy variables to prevent potentially ableist interpretations of the language. The notion of “reference coding” also makes it clearer that a specific group is the reference to which all others are being compared.

RESULTS

In general, p values should be reported to at least 3 decimal places.

What was the critical value of Cook’s distance that was used to identify outliers?

What were the effect sizes for the various F statistics that were computed?

The lack of significant results in the EMG analyses may result from the covariates being uncentered and the suboptimal windows used to score data.

The subtype analyses appear inappropriate given that the subtypes were created across all participants in the dataset, yet only those in the high psychopathy were divided this way. It would be more appropriate to maintain the splits across intermediate and low groups as well.

DISCUSSION

Could the lack of surprise faces have been one reason for the lack of psychopathy-related results compared to previous meta-analyses?

I was surprised that no mention of the characteristics of the two groups of participants were discussed anywhere in the text. Was it only NEM on which they differed, or did they differ on other MPQ variables? And were the domains of the MPQ used or the primary trait scales?

Reviewer #2: This manuscript reports results from a study examining the relationship between psychopathic traits (as measured by PCL-R) and autonomic and facial mimicry responses to dynamic facial emotion expressions, in addition to emotion categorization accuracy and valance and intensity judgments. The idea for the study and the methodological design was impressive and would constitute an addition to the scant and inconsistent literature in this area. It appears that the authors largely found null results, at least for the analyses described in the manuscript, with the exception of significant results for the exploratory analyses where participants with high scores on the PCL-R (greater than or equal to 30) were divided into “high-NA” and “low-NA” subgroups through cluster analysis of the MPQ-BF scales. The methodology and study design are a strength of the manuscript, and the null results are interesting and worthwhile of discussion on their own. I commend the authors for clearly delineating between the a priori and post-hoc/exploratory analyses that they conducted. Unfortunately, the authors’ discussion of their results, particularly the questionable conclusions drawn from the data and insufficient justification provided for the chosen exploratory follow-up analyses, was a significant weakness of the manuscript. Based on the way that the statistical results are interpreted and described in the manuscript, it is not suitable for publication in its present form, as it does not meet criteria for publication (particularly, Criterion 4: “Conclusions are presented in an appropriate fashion and are supported by the data”). The merit of the study methodology is important, and should the authors revise the document to reflect the feedback below, the manuscript might be suitable for publication and constitute a valuable addition to the published literature in this area.

Major Concerns

(1) Questionable interpretation of statistical results.

a. In the abstract, discussion, and conclusion, the statistical results appear to be inappropriately described/interpreted.

b. For example, in the abstract, references are made to “trend level” results and a conclusion is stated that: “These findings provide marginal evidence for reduced intensity ratings of and sympathetic arousal to prototypical facial emotion expressions in high psychopathy individuals.” This is a generous interpretation of the findings at best, and verges on questionable practices at worst.

1. The findings for reduced intensity ratings did not survive removal of outliers. Per Figure 2, it is notable that one outlier on the very low end of psychopathy scores had an averaged intensity rating of 6 out of 6 across emotions. The results of this analysis with outliers included are reported as significant, and then reported as “trend level” once the outliers were removed and the p-value became greater than 0.05. If it is meaningful to include the results with outliers not removed, then the authors will need to explain the potential impact of the extreme outliers (particularly the one described on the low end of psychopathy scores) on the results.

2. The findings for reduced SCR were not significant when covariates were included. In the Data Analysis section (lines 189-195), it specifically states that the covariates of age and anxiety have been found to impact SCR, hence the inclusion of those variables as covariates. If the results of the analysis without covariates is meaningful, the authors need to provide an explanation of the impact that including vs. excluding those covariates may have had on the results.

2) Insufficient justification for follow-up exploratory analyses.

a. First, justification will need to be provided for the choice to conduct exploratory follow-up analyses on the “trend level” results in the first place. Why not just describe the null results as null results and leave it there? Null results are still meaningful and important, particularly when the study methods appear rigorous.

b. Second, justification will need to be provided for the choice of exploratory follow-up analyses. Specifically, why look at low and high NA? Beyond the fact that such subtyping has been completed in prior studies, why is this particular break-down supported by theory and expected to be related to SCR and emotion intensity ratings?

1. Table S2 in the Supplemental Materials shows that SCR amplitude was negatively correlated (r = -.024, p < .05) with both PCL-R Total score and PCL-R Factor 1 (which is interesting, considering the interpersonal/affective factor would be expected to be related to arousal to affective stimuli). Table S7 shows that Factor 1 did not show significant relationships with SCR amplitude, especially when corrected for multiple testing, but what about looking at the facet level (i.e., interpersonal vs. affective vs. impulsive vs. antisocial facets)? The authors need to explain why/how they decided on the high vs. low-NA psychopathy subtyping rather than other potential post-hoc analyses that could have been completed and may have theoretical backing.

2. The authors will also need to explain why they looked at high and low-NA only in the “high psychopathy” group rather than looking at high and low-NA at all levels of psychopathic traits and must discuss the impact that the choice may have had on the results. For example, how can the reader tell whether the findings of reduced SCR in the low-NA high psychopathy group are not related to low-NA more broadly?

i. In drawing conclusions from these findings, it would be helpful for the authors to explain to the reader what the low and high-NA clusters are (as constructs) when anxiety is controlled for as a covariate, particularly when interpreting the results shown in in Table 3 (p. 15).

Minor Concerns

(1) Psychopathy appears to be largely presented as a unitary construct through the reliance on Total PCL-R score and limited discussions of the factor and facet structure. Adding some more discussion about factor/facet-level results (and maybe the relationship between the factors/facets and the high-NA and low-NA subtypes) would improve the paper. From the references, it is clear that the low- and high-NA subtypes have shown associations with aspects of “primary” and “secondary” psychopathy. Adding information like that to the text would help contextualize these subtypes to the reader and would tie it in with the broader literature on psychopathy.

(2) The write-up is light on discussion of theory and appears to rely on “novelty” to support the publication of the study and its findings, which is not a sufficient reason to publish anything. Just because something is new does not necessarily mean that it is worthwhile. On the other hand, the theoretical basis, and the role of this study as a “theory test” to move the field forward would be far more compelling.

a. Regarding theory-based hypotheses, the hypothesis about valence ratings (based on one study done in an all-female sample) is a bit weak, particularly since you have an all-male sample. Obviously don’t change up your hypotheses post-hoc, but it is a point of illustration. You could have made that hypothesis based on theory, and it would have been a stronger one.

(3) Group Ns should be included in all tables/figures consistently.

(4) Where is the power analysis? The group sizes are small, especially for the subgroup analyses with low- and high-NA. The small group size is briefly mentioned as a limitation (p. 21) but mention of statistical power should be presented along with the results (although, see my concern above that the choice to do the exploratory analyses on “trends” in data).

(5) Further, where are the effect sizes? It is laudable that the authors plan to make the data available upon publication, but the reader should not have to run the analyses themselves in order to determine the effect sizes for these “trends.”

6. PLOS authors have the option to publish the peer review history of their article (what does this mean?). If published, this will include your full peer review and any attached files.

Reviewer #1: **Yes: **Stephen D. Benning

Reviewer #2: No

---

## [Author Response · Author response to Decision Letter 0]

3 Apr 2022

We have attached a document with our responses to reviewer and editor comments.

---

## [Decision Letter · Decision Letter 1]

20 May 2022

PONE-D-21-38079R1An examination of autonomic and facial responses to prototypical facial emotion expressions in psychopathyPLOS ONE

Dear Dr. Deming,

Thank you for submitting your manuscript to PLOS ONE. After careful consideration, we feel that it has merit but does not fully meet PLOS ONE’s publication criteria as it currently stands. Therefore, we invite you to submit a revised version of the manuscript that addresses the points raised during the review process.

Please address the comments raised by the reviewers below. In particular, please address the first comment from Reviewer #2 regarding clarification of any analysis (and associated hypotheses) that have been carried out post-hoc.

We look forward to receiving your revised manuscript.

Kind regards,

Hugh Cowley

Senior Editor

PLOS ONE

Reviewers' comments:

Reviewer's Responses to Questions

**Comments to the Author**

1. If the authors have adequately addressed your comments raised in a previous round of review and you feel that this manuscript is now acceptable for publication, you may indicate that here to bypass the “Comments to the Author” section, enter your conflict of interest statement in the “Confidential to Editor” section, and submit your "Accept" recommendation.

Reviewer #1: (No Response)

Reviewer #2: (No Response)

2. Is the manuscript technically sound, and do the data support the conclusions?

Reviewer #1: Yes

Reviewer #2: Yes

3. Has the statistical analysis been performed appropriately and rigorously? 

Reviewer #1: Yes

Reviewer #2: I Don't Know

4. Have the authors made all data underlying the findings in their manuscript fully available?

Reviewer #1: Yes

Reviewer #2: Yes

5. Is the manuscript presented in an intelligible fashion and written in standard English?

Reviewer #1: Yes

Reviewer #2: Yes

6. Review Comments to the Author

Reviewer #1: In reading over this revision, I am impressed with the authors’ attention to the details of the reviewers’ comments. In all, this work provides an important multimodal assessment of reactivity to facial emotional displays in a sample of incarcerated men using multiple methods of displaying facial expressions and various methods of measuring reactivity. It also then shows how psychopathy is not associated with these forms of reactivity. I have a few remaining clarifying questions for this submission.

Within the introduction, the literature summary on page 4, paragraph 2 made me wonder the degree to which reactivity to facial expressions differs in incarcerated samples relative to non-incarcerated samples broadly speaking. However, this is more a point of curiosity that arises based on the overall null results that might contrast against those obtained in non-incarcerated samples. Wilson et al. (2011) suggests there are no meaningful differences between non-incarcerated and incarcerated samples, whereas Dawes et al. (2012) included only two non-incarcerated samples, making that kind of moderator analysis impossible to perform.

In the method, it would be helpful to add a sentence describing precisely the evidence of 60 Hz line noise that required notch filtering. For example, a diagram of a prototypical trial with visible 60 Hz noise or the output of a fast Fourier transform showing a peak at 60 Hz could be added to the supplemental materials, or a verbal description of either of these results in the method could do the same.

When describing in the discussion that a relatively high impedance is a limitation, it would be important to do more than just state it as such. There should be a sentence or two describing why it is a limitation to the study (e.g., impedances that high may prevent reading the underlying signal; they may also inject noise into the data).

Reviewer #2: This manuscript reports results from a study examining the relationship between psychopathic traits (as measured by PCL-R) and autonomic and facial mimicry responses to dynamic facial emotion expressions, in addition to emotion categorization accuracy and valance and intensity judgments. Despite the null findings for most analyses described in the manuscript, the idea for the study and the methodological design is impressive and will constitute an addition to the scant and inconsistent literature in this area. The methodology and study design are a strength of the manuscript, and the null results are interesting and worthwhile of discussion. The changes made to the manuscript in response to editor and reviewer comments on the initial submission have brought the document into alignment with the criteria for publication. I have provided a few minor revision suggestions for the authors, with the first point (transparency about a priori and post-hoc data analysis decisions) being the most essential to address prior to publication.

1) Since the analyses were changed in response to reviewer/editor feedback, transparency is essential. It will be important to include some mention that the analyses (and thus, associated hypotheses) are post-hoc.

2) The authors use the word “neurotypical” throughout the manuscript (e.g., p. 5, lines 108 and 110), which appears to be a new addition in this revision. It seems that this word is used to refer to individuals who are not high in psychopathic traits; however, “neurotypical” can be interpreted in different ways, so it would be useful to briefly describe what is meant by this word in the authors’ manuscript, in particular.

3) On page 5, lines, 109-111, the authors include the following statement that could use a brief elaboration: “Studying high psychopathy people could also advance our understanding of how neurotypical individuals process facial expressions.” Adding a reason why this is the case would help the reader understand the authors’ reasoning and viewpoint.

4) The use of the phrase “high psychopathy people” (e.g., p. 5, lines 109-110) is a bit odd, and saying “individuals high in psychopathic traits” or some other wording might be better.

5) In the Discussion section (p. 24, lines 519-523), the authors mention that their null findings for facial mimicry (even in those low in psychopathic traits) were unexpected, as a past study in a non-incarcerated sample did find spontaneous facial mimicry. It would be helpful to briefly discuss how differences in incarcerated vs. non-incarcerated samples might play a role in the different results.

7. PLOS authors have the option to publish the peer review history of their article (what does this mean?). If published, this will include your full peer review and any attached files.

Reviewer #1: **Yes: **Stephen D. Benning

Reviewer #2: No

---

## [Author Response · Author response to Decision Letter 1]

25 May 2022

Included in this submission is a document responding to each of the comments we received.

---

## [Editor Report · Decision Letter 2]

16 Jun 2022

An examination of autonomic and facial responses to prototypical facial emotion expressions in psychopathy

PONE-D-21-38079R2

Dear Dr. Deming,

We’re pleased to inform you that your manuscript has been judged scientifically suitable for publication and will be formally accepted for publication once it meets all outstanding technical requirements.

Kind regards,

Stephen Benning

Guest Editor

PLOS ONE

Additional Editor Comments (optional):

Before being asked to serve as Guest Academic Editor, I was Reviewer 1 on this paper. Having read through the revisions that the authors submitted in response to the last round of reviews, I am satisfied that this manuscript:

1. Presents the results of original research

2. That have not been published elsewhere

3. With experiments, statistics, and other analyses are performed to a high technical standard and are described in sufficient detail

4. And conclusions are presented in an appropriate fashion and are supported by the data, including important null results for the facial mimicry and psychopathy fields.

5. The article is presented in an intelligible fashion and is written in standard English, and

6. The research meets all applicable standards for the ethics of experimentation and research integrity.

Consistent with journal standards, this acceptance is contingent upon depositing at least the minimal data set in a repository or as supplemental materials (as described here: https://journals.plos.org/plosone/s/data-availability) and either providing the link to that repository or uploading the files with the manuscript so that:

7. The article adheres to appropriate reporting guidelines and community standards for data availability.

Your cover responses indicated that these data would be available via Figshare, so I presume the link there just needs to be revealed. Congratulations on your fine work! The location of the data in Figshare can be given on the title page if other arrangements are not possible.

---

## [Editor Report · Acceptance letter]

24 Jun 2022

PONE-D-21-38079R2 

An examination of autonomic and facial responses to prototypical facial emotion expressions in psychopathy 

Dear Dr. Deming:

I'm pleased to inform you that your manuscript has been deemed suitable for publication in PLOS ONE. Congratulations! Your manuscript is now with our production department. 

Kind regards, 

on behalf of

Dr. Stephen Benning 

Guest Editor

PLOS ONE